# Biocomposite Foams with Multimodal Cellular Structures Based on Cork Granulates and Microwave Processed Egg White Proteins

**DOI:** 10.3390/ma16083063

**Published:** 2023-04-13

**Authors:** Giorgio Luciano, Adriano Vignali, Maurizio Vignolo, Roberto Utzeri, Fabio Bertini, Salvatore Iannace

**Affiliations:** Institute of Chemical Sciences and Technologies (SCITEC) “Giulio Natta”, National Research Council (CNR), Via Alfonso Corti 12, 20133 Milan, Italy; adriano.vignali@scitec.cnr.it (A.V.); maurizio.vignolo@scitec.cnr.it (M.V.); roberto.utzeri@scitec.cnr.it (R.U.); fabio.bertini@scitec.cnr.it (F.B.);

**Keywords:** agricultural waste, egg white proteins, cork, foaming, microwave, flame resistance, insulating panels

## Abstract

In an effort to reduce greenhouse gas emission, reduce the consumption of natural resources, and increase the sustainability of biocomposite foams, the present study focuses on the recycling of cork processing waste for the production of lightweight, non-structural, fireproof thermal and acoustic insulating panels. Egg white proteins (EWP) were used as a matrix model to introduce an open cell structure via a simple and energy-efficient microwave foaming process. Samples with different compositions (ratio of EWP and cork) and additives (eggshells and inorganic intumescent fillers) were prepared with the aim of correlating composition, cellular structures, flame resistance, and mechanical properties.

## 1. Introduction

Cork is a light, renewable, and biodegradable material which is extracted from the bark of the cork oak tree (*Quercus suber* L). The world’s cork production is estimated at around 200 ktons per year, which, considering the density of the cork (120–240 kg/m^3^) [1], means about 1 Mm^3^ per year; 80% is produced Portugal and Spain, while the other 20% is produced in Morocco, Algeria, Tunisia, France, and Itay [2]. It is used in a variety of products, from construction materials to gaskets, and, most importantly, as a stopper for premium wines. A by-product resulting from the transformation processes are cork granulates, defined as cork waste. It comes from the cutting stage and the material rejected at the selection stage during cork stopper production. Cork granulates can be used to produce (a) “expanded cork agglomerates” through the agglutination of granulates without using any synthetic agents to bind or (b) “composition cork” through the binding of cork granulates with natural or synthetic resins.

The possibility of reusing cork by-products for applications in the construction industry has been widely studied [3]. Due to their homogeneous closed cellular structure, they are used for thermal and acoustic insulation in building construction [4,5,6,7] and for the preparation of lightweight wood-based sandwich structures [8]. The mechanical and thermal properties of cork panels depend upon the characteristics of the granulated cork (gradation, density, size, and grain size distribution) and on the type of the binder used, usually based on urethane, melamine, or phenolic resins. Recently, biobased binders and adhesives such as, for example, the resin of cork (suberin) [9], beeswax, rosin [10], and chitosan [11] are receiving increasing attention due to interest in reducing the carbon footprint of insulating products through the development of fully biobased agglomerated cork panels.

Proteins from plants (soy, wheat, maize, or rapeseed) and animal sources (casein, blood) and their use as wood adhesives have been investigated for a long time, but their applications are still limited due to their poor properties in wet conditions. For these reasons, proteins need to be physically or chemically modified or used with a crosslinker to provide sufficient cohesive strength to withstand wet conditions [12].

Egg white proteins (EWP) possess unique stabilizing properties in food foams, and their widespread traditional uses include, for example, meringues, mousses, and baked goods [13]. They are also a low-cost, versatile, readily available, and highly effective functional additive. They have a long history of application in medicine and construction, and, starting from the 20th century, their applications include paper, leather, textiles, medicine, the chemical industry, ceramics, plastics, paints, and more. Physical mixing and chemical cross-linking of egg white with other materials has been explored to make degradable packaging films, bioceramics, bioplastics, biomimetic films, hydrogels, 3D scaffolds, bone regeneration, biopatterning, and biosensors [14]. Albumin, the main protein of egg white, has been very recently studied as a model protein for wood adhesives [15].

The scope of the paper was to study the foaming properties of egg white proteins in the presence of cork granulates in order to manufacture fireproof, lightweight insulating panels with a multimodal open/closed cell structure. The EWP open cell structure was introduced with the aim of potentially improving the sound absorption properties together with both the thermal and sound insulation properties given by the closed cellular structure of cork granulates.

To increase the sustainability of the manufacturing process, we employed microwave (MW) heating technology as the source of heating. Among other advantages, MW heating is a well-established technology employed in the food industry, and its interaction with egg proteins is widely studied [16]. Its application in materials science has emerged over the years, with many innovative technologies exponentially increasing due to its unique characteristics of lower processing time and lower energy consumption [17]. The biocomposite foams were characterized by means of an electronic microscope and energy dispersive X-ray (SEM-EDX), Fourier-transform infrared spectroscopy (FTIR), thermogravimetric (TG) analysis, mechanical compression, and flammability tests.

## 2. Materials and Methods

Egg white from “Le Naturelle“ (1 kg brick) eggs (cat A and B), were bought from a local market. Cork was purchased from Kork-deco.de (1–3 mm particle size).

Al(OH)_3_ powder, commonly used as a flame retardant, was prepared by mixing a 1 M solution of NaOH (reagent grade from Merck (Darmstadt, Germany) and a stoichiometric quantity of Al_2_O_3_ (99.8%). The precipitate, rinsed by deionized water, was filtered (Whatman 1 CHR) and then put in oven at 80 °C for 12 h.

For biocomposite foam preparation, a rectangular silicon mold (25 cm × 5 cm × 12 cm in size) was used. A typical preparation for a brick was performed by vigorously mixing 450 g of egg white and 150 g of cork granulates using an egg whisk in order to introduce air bubbles in the protein solution. Even if several different mixers were tested (i.e., crushing mixer, cement mixer, etc.), the manual egg whisk beater was the best performer. It was able to not damage the cork grains, due to the absence of blades, during the mixing process and to foam the egg white at the same time. EWP foams are generated by two mechanisms: (i) the incorporation of air during the initial mixing and (ii) the generation of water vapor during the microwave heating.

After the mixing process, depending on the sample composition, we added Al(OH)_3_ and/or eggshells that were previously crushed using a mortar to reach a final grain dimension of 1–2 mm or less. Eggshells were added to the formulation as a CaCO_3_-based filler.

The final mixture was foamed in a microwave oven (Samsung (Seoul, South Korea) M/O 20LT GE71A) with 20 L of volume in the internal space and operating at 2.45 GHz frequency. The heat treatment, according to the dimension of the samples, was 5 min at 350 W (first stage), followed by a drying stage of 20 min at 150 W until the weight remained unchanged. Infrared photos were taken with SEEK (Santa Barbara, CA, USA) Shotpro Thermal Camera during the MW foaming. At the end of the first stage, the temperatures on the surface of the foamed samples were in the range of 90–110 °C (Figure 1).

The compositions of the biocomposite foams are reported in Table 1 in terms of weight ratio, considering cork weight as 1.

The morphology and elemental analysis of the foamed composite materials were performed with a scanning electron microscope (SEM) equipped with a probe for energy dispersive X-ray analysis (EDX). In detail, it is a HITACHI (Tokyo, Japan) TM3000 benchtop SEM (15 kV). SEM analysis was carried out on cross-section obtained by sawing the samples with a razor blade.

FTIR spectra were recorded using a Perkin Elmer (Waltham, MA, USA) FTIR Spectrum Two spectrophotometer. FTIR spectra were acquired in attenuated total reflection (ATR) mode in the range of 4000–400 cm^−1^ (16 scans, resolution 8 cm^−1^).

Thermogravimetric analysis (TGA) was performed with a TGA 8000 Perkin-Elmer analyzer operating at a constant heating rate of 20 °C/min. TGA data and derivative thermogravimetry (DTG) were carried out both in a nitrogen atmosphere and in an oxygen one, in the temperature range between 40 °C and 700 °C.

The density of the materials was evaluated by determining the mass with a precision balance and measuring the volume with a caliper for each sample. Mechanical properties of the materials were measured by means of uniaxial compression tests performed with a Zwick-Roell (Ulm, Germany) Z010 mechanical testing machine with a load cell of 2.5 kN. Parallelepiped-shape specimens (height of 20 mm, length of 30 mm, and width of 30 mm) blade-cut from bricks were compressed to 70% of their initial height at 10 mm/min. The reported data of density and mechanical properties were measured and averaged on three specimens for each sample.

Flame resistance of the composites was tested by exposing samples to a flame, and the affected area was photographed and quantified. This test was performed based on the EN ISO 11925-2 [18]: the samples were placed on a metallic clamp at 15 cm from the ground and exposed to the torch flame near a border for 10 min [19].

## 3. Results and Discussion

### 3.1. The Foaming Process

As reported by Dong et al. [14], “Egg white’s unique processing characteristics, such as foaming, emulsifying, heating stability, gelling ability, adhesiveness and biocompatibility, it can be mixed with other materials and cross-linked to create novel synthetic materials”, which can be more widely studied in a wide range of applications, from bio-ceramics to micro/nano bio-patterning. In this paper, we used EWP as a matrix model to introduce an open cell structure via a simple and energy-efficient microwave foaming process, which is known to affect the functional properties of egg white when compared to traditional thermal treatments [20] (Figure 2).

Proteins are extensively used in aerated systems because they have both hydrophilic and hydrophobic groups. The hydrophilic groups are oriented towards the water phase, while the hydrophobic groups are orientated towards the air phase. Thus, they can efficiently absorb at the air–water interface. Egg white (EW) is a complex system of proteins that are able to form a continuous viscoelastic network; therefore, EWPs can form viscoelastic interfacial film and stable foam [21]. In this work, EWP foams were generated by incorporating air during the initial mixing. During microwave heating, the cellular structure is stabilized by the formation of protein aggregates that leads to the formation of a porous gel network [22]. The open cell structure is then formed during the following drying stage, where water is removed.

It has been demonstrated that several types of substances can be extracted from cork in presence of subcritical water (SCW). In particular, carbohydrate-rich extracts can be obtained at temperatures in the 120–200 °C range, mainly cork hemicellulose. In the same conditions, SCW can extract ca. 96% of the total content of phenolics while extracts composed of up to 36% phenolics were obtained at the lower temperature range of 50–120 °C [23]. In order to analyze the extracted substances in the presence of EWP, we have performed preliminary studies aimed to compare the extracts in the presence of both SCW and EWP. The cork extract solution was prepared, leaving the cork grains in the egg white protein solution for 20 min in order to simulate the manufacturing process of the samples of Table 1.

Figure 3a shows the visual aspect of the EWP and SCW before and after the extraction of the granulated cork substance. The FTIR spectra reported in Figure 3b–d are referred to as samples A, B, and C. While samples A and C were analyzed in the presence of water, for sample B, the analysis was performed on the dry solid that was extracted by drying the EWP solution. Water presents a broad band with a maximum at ~3400 cm^−1^ and a peak at 1643 cm^−1^ [24]. FTIR spectra of the dry solid containing both protein macromolecules and cork extract present several peaks that were observed in sample B, confirming the presence of substances extracted from cork. In particular, the peak at 1580 cm^−1^, assigned to C=C aromatic bond vibration, and the peak at 1050 cm^−1^, assigned to the C-OH vibration, can be ascribed to the presence of phenolics extracted from cork [23,25]. Egg white proteins presents many other peaks: 1082 cm^−1^ assigned to -C-O stretching, a peak at 1158 cm^−1^ assigned to -C-O (stretching), -CH_2_- (bending); 1235 cm^−1^ -C-O (stretching), -CH_2_- (bending), 1458 cm^−1^ -C-H(CH_2,_CH_3_) bending (scissoring) ~1546 cm^−1^ to -N-H (bending), -C-N (stretching), 1590–1720 cm^−1^ -C=O, -C-N stretching, 2851 cm^−1^ -C-H(CH_2_) stretching (symmetry), and at 2920 cm^−1^ -C-H(CH_2_) stretching (asymmetry) [26].

### 3.2. Foam Morphology

Figure 4 shows the SEM images of selected foamed samples. The micrographs (a) and (b) refer to the sample PDSB3, which is the biocomposite foam with the highest content of EWP and both inorganic fillers. They clearly show the presence of open cells connecting the cork granules; the latter are characterized by a typical closed cell structure [1,27]. A similar cellular structure was observed in sample PDSB2, which contains the same EWP/cork weight ratio and a lower amount of inorganic filler in comparison to the sample PDSB3. When the content of EWP is lower, such as in samples PDSB1, PDSB4, and PDSB5, the amount of the open cell structure is reduced, and many cork granules appear to be bonded by thin protein layers (Figure 4c,d). A large number of closed cells of the cork granules are clearly embedded by the EWP matrix (Figure 4c), and this is evidence of the good binding properties of the protein phase in the biocomposite foams.

The presence of inorganic fillers in the cell walls of the EWP phase can be depicted in Figure 4a,b. With the aim of investigating the particle distribution of the inorganic fillers in the foam, EDX analysis (Figure 4f) was performed on the sample PDSB3 at high magnification (Figure 4e). At this magnification, Al(OH)_3_ appears to be homogenously distributed in the matrix phase in which the cork granules are embedded.

As discussed below, the good distribution of aluminum trihydrate in the EWP phase contributed to the fire resistance of the bio-composite foams.

### 3.3. Density and Mechanical Properties of Foams

The density (*d*) and the mechanical compression properties of the samples, namely the Young’s modulus (*E*) and the maximum compression strength (*σ_max_*), are reported in terms of mean values with standard deviations in Table 2.

All the samples present similar density values of about 300 kg/m^3^, except for PDSB1 (the sample prepared without eggshell and Al(OH)_3_), which shows a lower density (~200 kg/m^3^). The density of cork is in the range of 120–240 kg/m^3^ [1], and these values are consistent with the density of the sample PDSB1. The addition of inorganic fillers led to materials with higher densities.

The compression stress–strain (σ–ε) curves, shown in Figure 5, present two different types of mechanical behavior. In detail, the first series of samples (PDSB1, PDSB2, and PDSB3) exhibits, until 50% of strain, a broad area of σ−ε curves with low values of *E* (~1 MPa) and almost constant stress (<1 MPa). After that, a rapid increase in stress (*σ_max_* equal to 2–4 MPa) was observed as strain increased due to a partial cell densification. The stress–strain curves of the second series of samples (PDSB4 and PDSB5) exhibited a behavior similar to cork, characterized by an initial linear portion due to the elastic deformation of cells, where Young’s modulus was determined, followed by a yield and then by a progressive increase in the curve related to a partial densification. Moreover, the latter series of biocomposite foams is stiffer compared to the first one. In particular, PDSB4 and PDSB5 show values of *E* and *σ_max_* equal to 12 MPa and 4 MPa, respectively, and are close to the mechanical properties of cork, as evaluated in several studies [28,29,30]. These results can be ascribed to the different compositions and morphologies of biocomposite foams. Indeed, the higher amount of EWP in samples PDSB2 and PDSB3 led to softer materials with cork embedded in a large interconnecting open cell structure compared to samples PDSB4 and PDSB5. In the latter cases, the lower amount of protein led to biocomposite foams characterized by cork granulates partially packed and partially embedded in the EWP foam, as shown in Figure 4d. During the compression tests, the partially packed cork granulates contributed to the increase in the sample stiffness. The presence of inorganic fillers contributed to further increase the elastic modulus of these biocomposite foams.

### 3.4. Thermogravimetric Analysis

In Figure 6, we reported the thermogravimetric curves, under inert and oxidative atmospheres, of the sample composed of egg white and cork (PDSB1) and the sample with the added inorganic fillers (PDSB3). It can be noticed that all curves present a small mass loss for temperatures below 200 °C due to the moisture loss. Under an inert atmosphere (Figure 6a), the main degradation takes place in the temperature range between 250 °C and 550 °C and can be attributed to the decomposition of EWP [31] and cork components, such as hemicellulose, cellulose, suberin, and lignin [32]. DTG curves show the reaction rates during thermal degradation and are characterized by two main peaks centered at about 320 °C and 420 °C, respectively. The addition of inorganic fillers (CaCO_3_ and Al(OH)_3_) in both cases slows down the degradation process and therefore improves the thermal stability of the materials. The residue obtained at 700 °C for PDSB1 is ca. 30 wt.% due to the char formed from cork, while, in the case of PDSB3, the residue amount is about 60 wt.% due to the marked contribution of inorganic particles. The decomposition process under an oxidative atmosphere can be divided into two regions (Figure 6b): a first mass loss between 50 °C and 150 °C, attributed to the water evaporation; followed by a two-stage decomposition in the temperature range 200–450 °C related to the degradation of EWP and of lignocellulosic components contained in the cork; and a final weight loss attributed to the char combustion [32]. At 700 °C, PDSB1 shows a residue of about 0%, whereas PDSB3 presents a relevant residue (ca. 40 wt.%) due to the inorganic fillers present in the materials. The higher thermal stability of sample PDSB3 compared to PDSB1 was also confirmed under oxidative conditions, pointing out the positive effect of the inorganic filler in delaying the degradation process.

### 3.5. Flammability Tests

Figure 7 shows the set-up adopted for testing the flame resistance of selected samples according to reference [19]. Infrared images were taken during the burning phase in order to measure the temperature distribution around the burning area. For comparison, a commercial wood conglomerate panel was also analyzed. As a first observation, it is worth noting that the area having temperatures in the range of 250–500 °C, where most of the thermal degradation phenomena occurred (see Figure 6), is lower in the biocomposite foams with respect to the commercial wooden panel.

Both the commercial and the biocomposite samples showed self-extinguishing behavior. However, the fire mechanism, which includes ignition, combustion, fire propagation, and extinction, were different. In particular, for both the commercial panel and the sample PDSB1, we observed the formation of a burning flame on the surface, while no flame was observed for the sample PDSB3. As expected, the latter biocomposite foam showed the best fire resistance due to the presence of both inorganic fillers (CaCO_3_ and Al(OH)_3_). Pictures of the tested samples before and after the test are reported in Figure 7 and Figure 8.

The absence of an incandescent area in the PDSB3 samples (Figure 7e) can be ascribed to the different roles of the inorganic fillers. In particular, at high temperatures, the aluminum idroxide, which is the most frequently employed flame retardant in polymers, is converted to aluminum oxide and water molecules. This activity interrupts the ignition process with two different mechanisms: (i) the endothermic process absorbs heat, and the polymer and the flame become colder (heat sink), (ii) the produced water vapor dilutes the combustible gaseous fuel. In general, these additives are employed in plastics in very high concentrations, usually more than 50% by weight. Therefore, negative effects on the material’s physical and mechanical characteristics are often observed [33]. In the case of the sample PDSB3, the total amount of inorganic fillers is 40 wt.%, with 13 wt.% Al(OH)_3_ and 27 wt.% CaCO_3_; thus, the content of aluminum hydroxide is lower than the typical amount used in the polymer matrix. This very good flame resistance can be ascribed to the synergistic charring effect due to the cork, along with the presence of both of the inorganic fillers in a protein matrix. Therefore, the combination of the two selected fillers appears to be a valid approach to design and produces insulating panels with satisfactory fire-retardant behavior.

## 4. Conclusions

In this paper, we present a novel biocomposite foam with a multimodal cellular structure through the use of a simple and energy-efficient microwave (MW) foaming process. Egg white proteins were used as a model matrix to generate a foamed structure embedding cork granulates.

The analysis performed on the samples highlighted the following:(1)Biocomposite foams were characterized by a bimodal cell size distribution: an open cell EWP matrix with cell size between 100 and 500 µm cork and granulates with closed cells with a cell size in the range of 30–50 µm.(2)The mechanical properties of the biocomposites analyzed in this work were mainly affected by the volumetric fraction of the foamed matrix. The lower amount of EWP resulted in materials with a higher elastic modulus and compression strength due to the presence of cork granulates partially packed in the EWP matrix.(3)Thermal stability and flame resistance of the samples were improved by the addition of 13 wt.% of Al(OH)_3_ and 27 wt.% CaCO_3_. A synergistic charring effect, due to the cork along with the presence of both of the inorganic fillers in a protein matrix, was observed.

## Figures and Tables

**Figure 1 materials-16-03063-f001:**
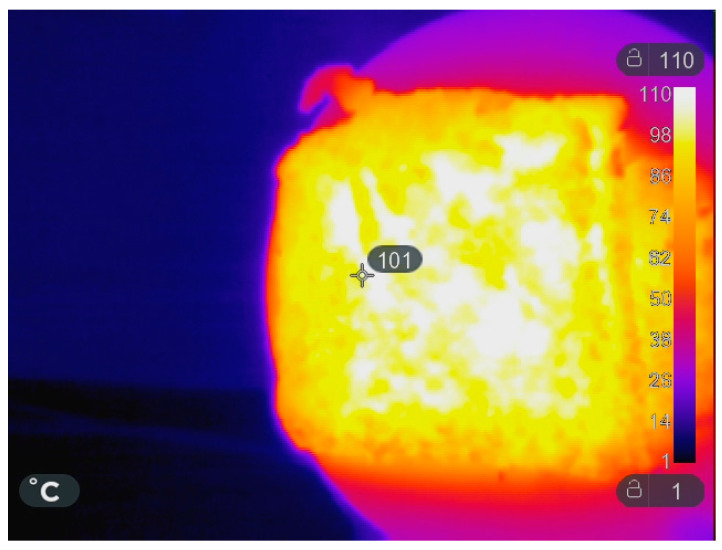
Thermal image of brick sample preparation at the end of the first stage of microwave heating.

**Figure 2 materials-16-03063-f002:**
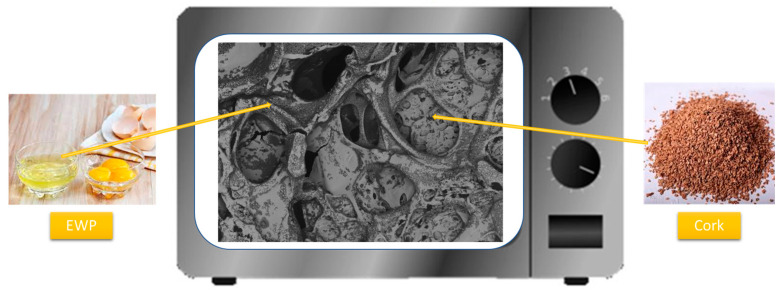
Schematic representation of the microwave foaming process of EWP in the presence of cork.

**Figure 3 materials-16-03063-f003:**
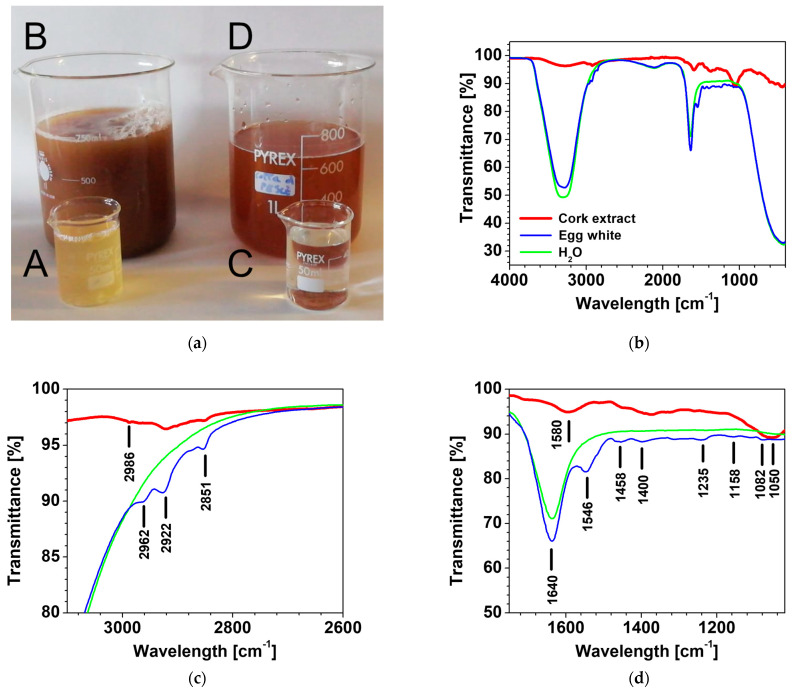
(**a**) EWP before (A) and after extraction (B); SCW before (C) and after extraction (D); (**b**–**d**) FTIR spectra.

**Figure 4 materials-16-03063-f004:**
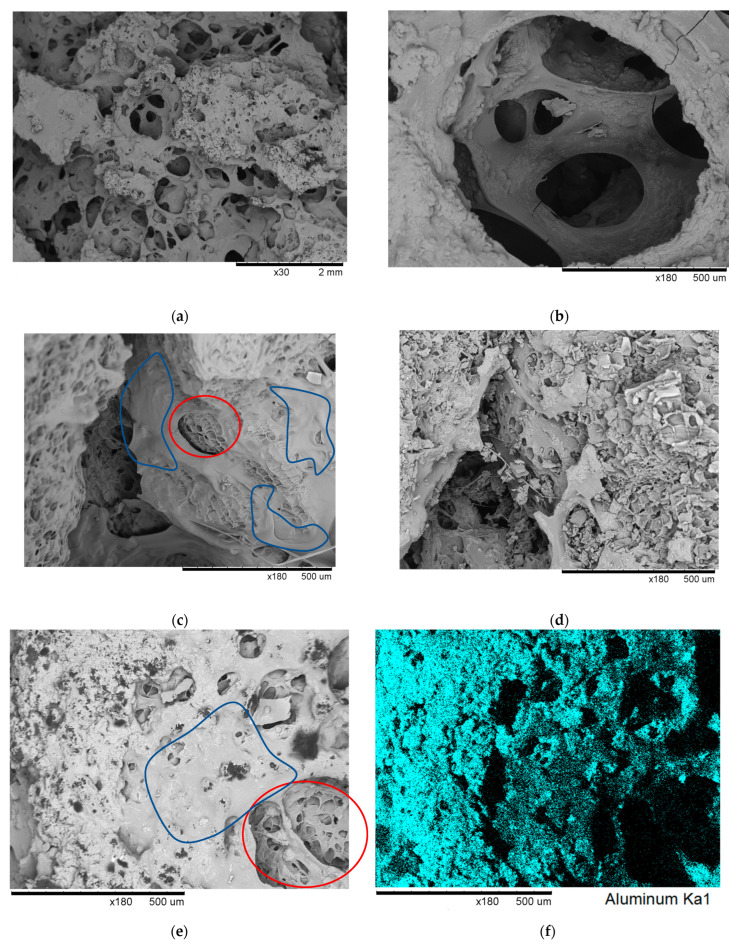
SEM micrographs of samples (**a**,**b**,**e**) PDSB3, (**c**) PDSB1, and (**d**) PDSB4; (**f**) aluminum mapping of sample PDSB3 by EDX analysis. In the micrographs, the red areas contain cork grains, and the blue areas are richer in egg white.

**Figure 5 materials-16-03063-f005:**
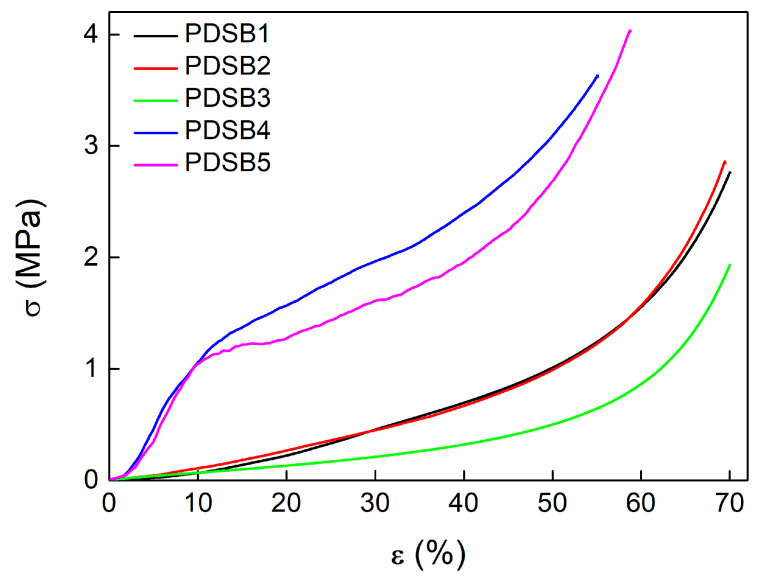
Stress–strain curves for compression-tested samples.

**Figure 6 materials-16-03063-f006:**
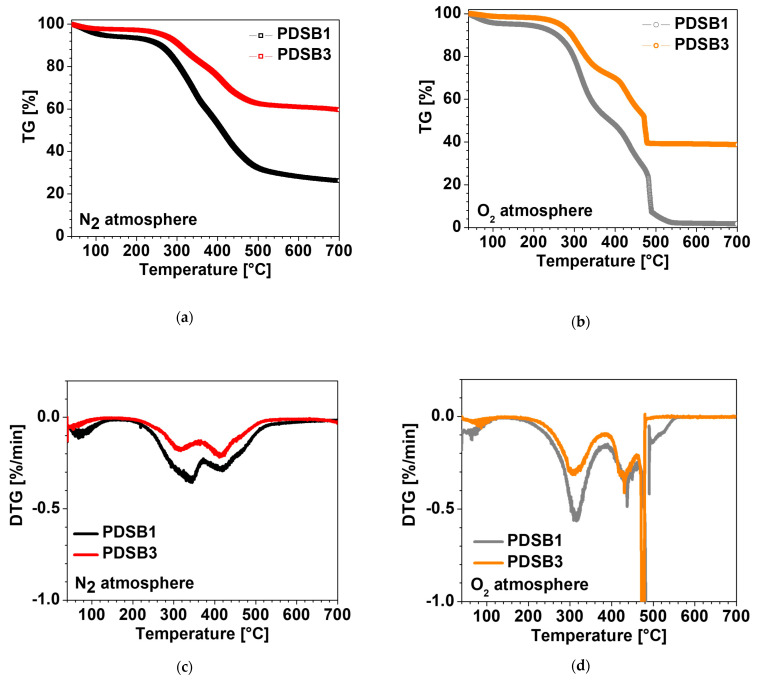
TGA thermograms of samples PDSB1 and PDSB3 under (**a**) N_2_ atmosphere and (**b**) O_2_ atmosphere; (**c**,**d**) are the respective graphs of the DTG analysis.

**Figure 7 materials-16-03063-f007:**
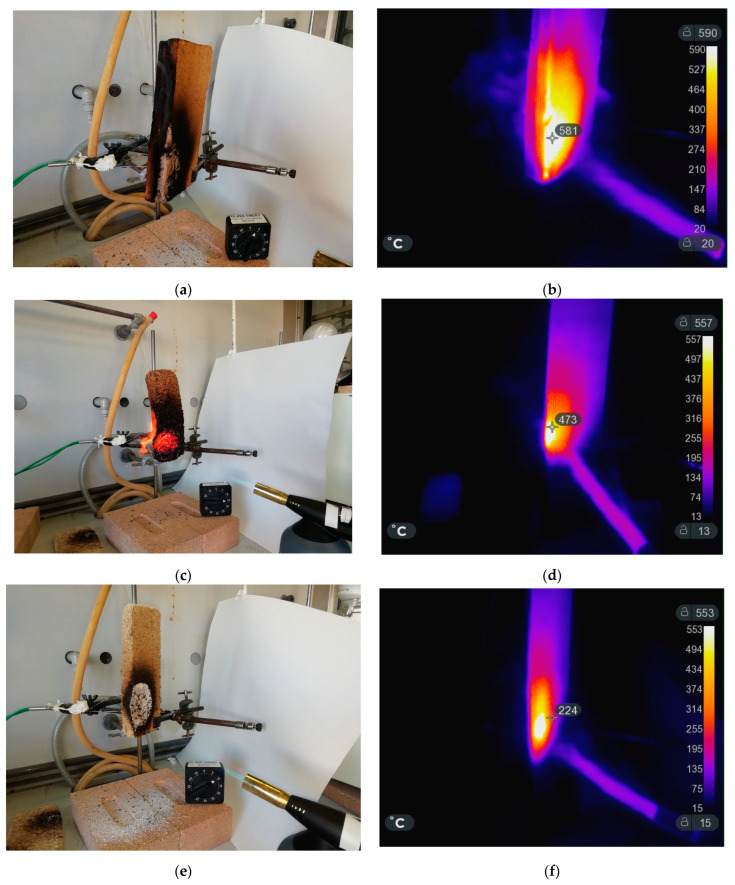
The experimental set-up and thermal images during flame resistance tests on (**a**,**b**) commercial wood conglomerate panel, (**c**,**d**) PDSB1 and (**e**,**f**) PDSB3.

**Figure 8 materials-16-03063-f008:**
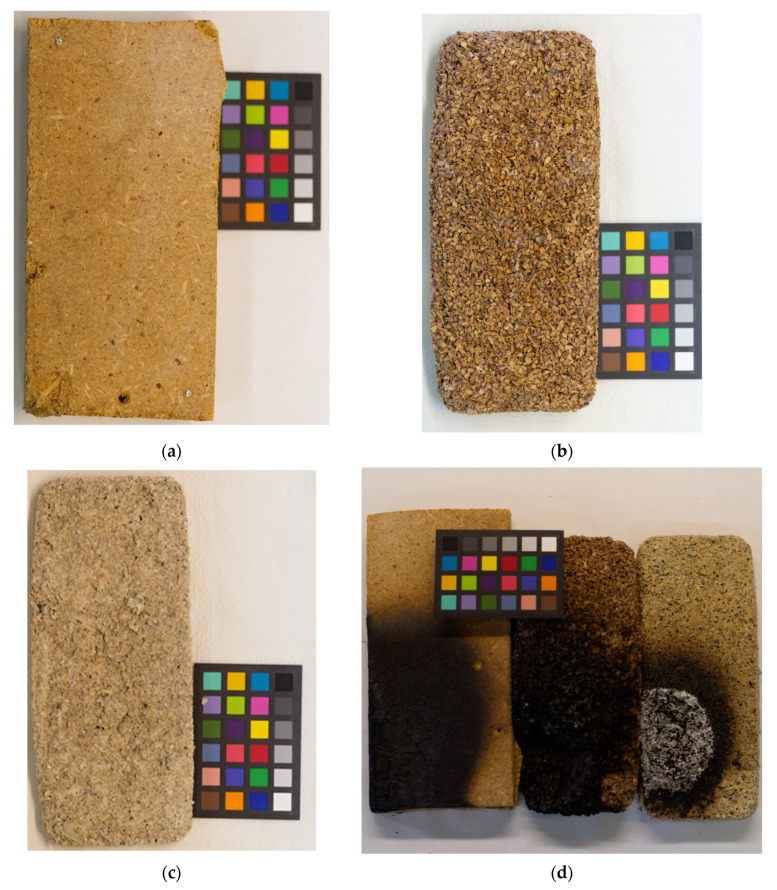
Samples analyzed: (**a**) commercial wood conglomerate panel, (**b**) PDSB1, (**c**) PDSB3, and (**d**) the three samples after 10 min of the flammability test.

**Table 1 materials-16-03063-t001:** Composition of the samples by weight ratio with respect to the amount of cork.

Sample	EWP	Egg Shells (CaCO_3_)	Al(OH)_3_	Cork
PDSB 1	3	-	-	1
PDSB 2	3.5	2	-	1
PDSB 3	3.5	2	1	1
PDSB 4	3	-	1	1
PDSB 5	3	2	-	1

**Table 2 materials-16-03063-t002:** Density and mechanical properties for the biocomposite foams.

Sample	*d* (kg/m^3^)	*E* (MPa)	*σ_max_* (MPa)
PDSB1	204 ± 8	1.00 ± 0.17	2.73 ± 0.05
PDSB2	327 ± 26	1.27 ± 0.24	2.74 ± 0.13
PDSB3	365 ± 52	0.76 ± 0.27	1.96 ± 0.18
PDSB4	295 ± 15	12.30 ± 3.32	3.85 ± 0.37
PDSB5	364 ± 35	11.90 ± 3.38	3.94 ± 0.09

## Data Availability

Data available upon request.

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
