# Peer review of "Biocomposite Foams with Multimodal Cellular Structures Based on Cork Granulates and Microwave Processed Egg White Proteins"

_materials, 2023, doi:10.3390/ma16083063_

Round 1

Reviewer 1 Report

This paper provides a thorough investigation of Egg white proteins was used as a matrix model to introduce an open cell structure via a simple and energy-efficient microwave foaming process, The authors drew broad and significant conclusions through FTIR and SEM studies. However, there are a few areas that could be improved in this paper.

Firstly, it would be better to discuss the Amide I in FTIR part, because it’s an important factor to protein structure.

Secondly, the authors did not include stability performance of the protein product extracted through the microwave foaming process, could the author elaborate this part?

Reviewer 2 Report

The article is very innovative and the preparation process is simple. I have given some comments and suggest to accept it after modification.

1. Why does the author want to use egg to prepare foams? Because it is a common food, and recently because of influenza, it is difficult to buy eggs in many countries. Unless you prepare edible foams, I think it will have commercial value.

2. Line 90-91

When mixing with an egg whisk, what is the specific speed of the eggbeater so as not to destroy cork particles, and please indicate the model of the eggbeater.

3. Line 95

The foam is heated in an oven, but the preface (lines 70-71) mentions that it is heated by microwave. Please confirm the specific heating method.

4. In the second part, please explain what a foaming agent is and why Al(OH)3 is added.

5. When microwave foaming, how to ensure that the mixture will not be destroyed by boiling at high temperature?

6. What are the advantages of PDSB foam compared with other types of biomass foam?

7. What is the production cost of this PDSB foam
